# Responsible Innovation in Plant Breeding: The Case of Hybrid Potato Breeding

**DOI:** 10.3390/plants12091751

**Published:** 2023-04-24

**Authors:** Dirk Stemerding, Koen Beumer, Rosanne Edelenbosch, Jac. A. A. Swart, Michiel E. de Vries, Emily ter Steeg, Conny J. M. Almekinders, Pim Lindhout, Luuk C. M. van Dijk, Paul C. Struik

**Affiliations:** 1Independent Researcher Biotechnology and Society, 8012 EV Zwolle, The Netherlands; 2Copernicus Institute of Sustainable Development, Utrecht University, 3584 CS Utrecht, The Netherlands; k.beumer@uu.nl; 3Rathenau Instituut, 2593 HW The Hague, The Netherlands; r.edelenbosch@rathenau.nl; 4Energy and Sustainability Research Institute Groningen, University of Groningen, 9747 AG Groningen, The Netherlands; j.a.a.swart@rug.nl; 5Solynta, 6703 HA Wageningen, The Netherlands; michiel.devries@solynta.com (M.E.d.V.); pim.lindhout@solynta.com (P.L.); 6Development Economics, Wageningen University and Research, 6706 KN Wageningen, The Netherlands; emily.tersteeg@wur.nl; 7Knowledge, Technology and Innovation, Wageningen University and Research, 6700 EW Wageningen, The Netherlands; conny.almekinders@wur.nl; 8Centre for Crop Systems Analysis, Wageningen University and Research, 6708 PE Wageningen, The Netherlands; luuk.vandijk@wur.nl (L.C.M.v.D.); paul.struik@wur.nl (P.C.S.)

**Keywords:** hybrid potato, responsible innovation, food security, smallholder farming

## Abstract

As an emerging innovation, hybrid potato breeding raises high expectations about faster variety development and clean true potato seed as a new source of planting material. Hybrid breeding could, therefore, substantially contribute to global food security and other major sustainable development goals. However, its success will not only depend on the performance of hybrid potato in the field, but also on a range of complex and dynamic system conditions. This article is based on a multidisciplinary project in which we have studied the innovation dynamics of hybrid potato breeding and explored how these dynamics may shape the future of hybrid potato. Inspired by the approach of responsible innovation, we closely involved key players in the Dutch and international potato sector and other relevant actors in thinking about these potato futures. An important and recurrent theme in our work is the tension between the predominant commercial innovation dynamics in plant breeding and promises to respond to the global challenges of food security, agrobiodiversity and climate change. In this article, we, therefore, discuss responsible innovation strategies in (hybrid) potato breeding, which may help to bridge this tension and finally reflect on the implications for the field of plant breeding in general.

## 1. Introduction: The Prospect of Hybrid Potato Breeding

As the third global food crop, the potato (mainly *Solanum tuberosum* L.) is highly important for food security and income generation in many regions of the world. However, compared with other major food crops, improvement of potato yield through breeding efforts has been slow. In this context, hybrid diploid potato breeding is considered a particularly promising development that may significantly accelerate the process of crop improvement. By fixing and combining desirable traits in diploid potato inbred lines, a hybrid breeding strategy provides new opportunities to improve yield, disease and pest resistance and tuber quality and to rapidly respond to diverse and changing market needs [1,2]. Owing to its far more complex and unpredictable genetics, the traditional tetraploid potato breeding system is inherently slow in producing new varieties and hence less efficient. Moreover, instead of the clonally propagated seed stocks in traditional potato multiplication, hybrid varieties will become available as true potato seed. This seed is basically pathogen-free, as most of the contaminating pathogens that accumulate in successive generations of tubers are not true seed-borne, and it can be easily produced, transported and stored. True hybrid potato seed could thus serve as an abundant source of clean and high-quality starting material, in the form of seed, seedlings or seed tubers, and eliminate the lack of access to good planting material as one of the main challenges for potato cultivation in developing countries [3,4].

Breeders welcome the prospect of hybrid diploid potato breeding as a game-changing development that can help adapt potatoes to various soils, climates and agro-ecologies, and make them resistant to pests, diseases, heat and drought [5]. Indeed, the promises of hybrid potato breeding clearly resonate with a more general debate about the role of plant breeding in responding to the global challenges of food security, agrobiodiversity and climate change [6,7]. A recurrent theme in this debate is the tension between, on the one hand, the predominant innovation dynamics in which breeding primarily serves an agro-industrial model of development and, on the other hand, the global challenges of poverty, hunger and resource scarcity that are central to the UN Sustainable Development Goals (SDGs). The agro-industrial model has resulted in the establishment of an international formal seed system, with seed firms as commercial suppliers of improved and certified seeds. However, the commercial supply of these seeds mainly targets and benefits the more well-positioned and technologically advanced farmers in the world, while many of the farmers facing the challenges of food security and climate change are resource-poor smallholders who still strongly rely on farm-saved seeds. These farmers may often decide not to adopt commercially improved crop varieties, as they will see the purchase of costly high-quality seeds as an uncertain and risky investment. Innovations in breeding that become available through the formal, science-driven and commercial seed system, therefore, often fail to reach smallholder farmers [8,9]. To more fully understand the potential of hybrid diploid potato breeding, we thus have to take into account the diverse and complex socioeconomic and agroecological system conditions in which farmers are operating.

### The Pursuit of Responsible Innovation

In this article, we discuss the future of hybrid potato from the perspective of responsible innovation, providing an approach that may help breeders consider the wider societal implications of their work in ways that are anticipatory, inclusive and responsive. Our experiences and findings are the result of a multidisciplinary project (POTAREI, [10]), in which researchers from different scientific backgrounds—including crop science, social science and ethics—worked together on the question of how to create conditions for responsible innovation in hybrid potato breeding that benefit the productivity, sustainability and diversity of current potato production systems. In this project, we considered the future of hybrid potato breeding as both open and uncertain. This future, indeed, will depend not only on the ultimate performance of hybrid potato in the field, but also on a range of societal actors, conditions and developments that may steer breeding in various directions.

The POTAREI project was initiated in collaboration with one of the founders of the Dutch start-up company Solynta. In 2008, this company began working on hybrid potato breeding as a radically new approach. While taking Solynta’s innovation journey as our starting point, we focused in our POTAREI project on the wider system conditions in which hybrid potato breeding as innovation must gain a foothold. Our aim was to better understand the complex and uncertain ways in which hybrid potato breeding and these system conditions may shape each other, and to consider the tensions that may arise between these conditions and the goals and promises that breeders have in mind for the future of hybrid potato. Our work was inspired by the approach of responsible research and innovation (RRI), which aims to take into account the complexities and uncertainties of innovation, while seeking ways in which innovation can respond to major societal challenges [11]. In this context, we explored how the global challenges of food security, agrobiodiversity and climate change can be made key drivers in the development of hybrid potato breeding.

## 2. Story Outline and Source Materials

Our story consists of seven parts. It begins with an account of Solynta’s innovation journey in hybrid breeding as the starting point of our POTAREI project. In the second part, we further explain how our project has been inspired by the approach of RRI, aiming at anticipation, inclusion, reflexivity and responsiveness. In the next four parts, we discuss the activities that we have been undertaking in our project and the major findings and lessons that resulted from these activities. In these parts, we show how we have translated the aims of responsible innovation in (1) a future-oriented exploration of hybrid potato breeding, including discussions of (2) the governance of potato genetic resources, (3) the role of hybrid potato breeding in responding to the UN Sustainable Development Goals and (4) responsible innovation strategies in (hybrid) potato breeding. Based on our major findings, we finally draw some more general conclusions about responsible innovation in plant breeding.

The content of this article is based in many ways on work that we published during and following the life of our project. Several parts of our story are taken directly from this work. In Box 1, we have included the major publications on which our story relies. A full list of project materials is available at: https://www.nwo.nl/projecten/313-99-301-0 (accessed on 19 April 2023).

Box 1Major publications from (or related to) our POTAREI project.2011Lindhout et al. Towards F1 hybrid seed potato breeding. *Potato Res. 54*, 301–312.2016De Vries et al. The potential of hybrid potato for East-Africa. *Open Agric. 1*, 151–156.2018Lindhout et al. Hybrid potato breeding for improved varieties. In *Achieving Sustainable Cultivation of Potatoes Volume 1 Breeding: Nutritional and Sensory Quality*; Wang-Pruski, G., Ed.; Burleigh Dodds Science Publishing Limited: Cambridge, UK, pp. 99–122.Beumer and Edelenbosch Hybrid potato breeding: A framework for mapping contested socio-technical futures. *Futures 109*, 227–239.2019Swart and Stemerding *Opportunities and challenges for hybrid potatoes in East-Africa*. International Workshop Report, University of Groningen, The Netherlands.2020Edelenbosch and Munnichs *Potatoes are the future. Three scenarios for hybrid potatoes and the global food supply*. Rathenau Instituut: The Hague, The Netherlands.Stemerding et al. *Potato futures: impact of hybrid varieties*. International Conference Report, NFP: The Hague, The Netherlands.2021Beumer et al. Innovation and the commons: lessons from the governance of genetic resources in potato breeding. *Agric. Hum. Values 38*, 525–539.Beumer and Stemerding A breeding consortium to realize the potential of hybrid diploid potato for food security. *Nat. Plants 7*, 1530–1532.Van Dijk et al. Hilling of transplanted seedlings from novel Hybrid True Potato Seeds does not enhance tuber yield but can affect tuber size distribution. *Potato Res. 64*, 353–374.2022Ter Steeg et al. Crucial factors for the feasibility of commercial hybrid breeding in food crops. *Nat. Plants 8*, 463–473.2023Struik et al. (Eds.) *The Impact of Hybrid Potato. The Future of Hybrid Potato from a Systems Perspective*; Wageningen Academic Publishers: Wageningen, The Netherlands, p. 183.Stemerding et al. Introduction; Chapter 1. In *The Impact of Hybrid Potato*; pp. 11–16.Lindhout and Struik, Hybrid potato breeding and production systems; Chapter 2. In *The Impact of Hybrid Potato*; pp. 17–30.Edelenbosch and Stemerding The future of hybrid potato: Responsible innovation with future scenarios; Chapter 3. In *The Impact of Hybrid Potato*; pp. 31–43.Swart and van de Poel Corporate social responsibility and hybrid potato breeding: Balancing economic, environmental and social challenges; Chapter 9. In *The Impact of Hybrid Potato*; pp. 141–160.Beumer and Almekinders Hybrid potato breeding for smallholder farmers in developing countries: Four models for public-private collaboration; Chapter 10. In *The Impact of Hybrid Potato*; pp. 161–174.Stemerding et al. Conclusion: Major findings and discussion; Chapter 11. In *The Impact of Hybrid Potato*; pp. 175–183.

## 3. Solynta’s Innovation Journey

Since its founding in 2008, Solynta has experimented with and tested the concept of hybrid breeding in potato. The company started to build a hybrid breeding program using diploid potato germplasm obtained from a pre-breeding program from Wageningen University and Research. To render this germplasm self-compatible, it was crossed with a wild potato species carrying a self-compatibility restorer gene *Sli*, which was made available by the Japanese researchers who identified this gene [1]. Thus, the company was able to initiate a hybrid potato breeding program by generating diploid homozygous inbred lines [4]. With the development of a sufficient number of highly inbred lines, it became possible to systematically combine genes and exploit heterosis in hybrid varieties, also introducing desirable traits such as disease resistance by marker-assisted introgression breeding [12,13,14]. In 2015, the first series of field trials were conducted in the Netherlands, testing the performance of experimental diploid potato hybrids grown from seedlings rather than seed tubers [3].

In the initial experimental phase of program development, Solynta worked on hybrid varieties and field testing with business development as its primary aim. However, whilst still unready for commercialization in highly developed potato markets, such as the Netherlands, the company started to consider the potential impact of hybrid potato on global food security. In 2016, the Solynta team undertook a fact-finding mission [15] and had the opportunity to test its genetic material in Sub-Saharan Africa, after being approached by a local organization (the Lake Albert Foundation) with a strong interest in hybrid potato seed. In this first trial, yields were recorded up to 29 t/ha, whilst average yields of local potato varieties ranged from 5 to 10 t/ha [3]. In low-income countries, the informal farm-based seed system has been identified as one of the main issues depressing yields, given a general lack of high-quality potato seed tubers and with potato seed degeneration as a serious threat [16]. It is well-known that farm-saved and locally multiplied seed tubers in East Africa are heavily contaminated by many diseases, some of them very difficult or even impossible to control [17,18].

From 2018 onwards, Solynta started building a team focused on various local partnerships with the aim of introducing hybrid potato in Sub-Saharan Africa. Hybrid varieties were tested in collaboration with different stakeholders in the potato value chain. Farmer trials showed that cultivation systems in the region may pose serious challenges. Potato smallholder farmers may have difficulties coping with vulnerable seedlings, requiring irrigation, careful soil management and intensive weed control, and may thus have to rely on (commercial) horticulture farmers to produce seed tubers from seedlings, so-called seedling tubers. Nonetheless, the commercial availability of more and cleaner potato starting material can result in general benefits: raising yields and strengthening local food security. Innovative seed-to-seed tuber multiplication systems are seen as a crucial step toward this goal [19]. Moreover, contrary to other hybrid seeds, hybrid potato seed can still be multiplied vegetatively by farmers, and hence, it is expected to result in (rapid) diffusion through both the formal and informal seed system. Currently, Solynta is registering hybrid varieties in several countries in Sub-Saharan Africa. The company aims at market-based solutions for food security challenges, and its continued focus on this region is based on commercial potential as well as potential impact.

Meanwhile, the Solynta breeding team made great progress in developing improved varieties. In 2017, the company succeeded in introducing a double resistance against *Phytophthora infestans* (the causal agent of late blight) in experimental hybrids through marker-assisted breeding technology within two years [12]. In 2016, the performance of experimental hybrids was shown to be comparable to some tetraploid checks in Europe, applying conventional potato agronomical practices [20]. Moreover, on the basis of more detailed knowledge of diploid potato genetics, it will be possible to further exploit the optimal combining abilities of available parental lines and develop efficient crossing strategies in hybrid breeding [21,22]. With the envisioned shift in potato starting material from conventional tubers to hybrid seed, the cropping system for hybrid potato also became an important subject of investigation. Different routes from true seed to ware potato are currently explored in agronomic research, which is performed by the company in collaboration with researchers from Wageningen University and Research, focusing on the optimal cultivation practices for crops raised from seedlings under (Dutch) field conditions [23,24,25].

During its innovation journey, Solynta has continuously been seeking to enlarge its research capacity through collaboration with academic partners, providing them with access to proprietary genetic material for (non-commercial) research [26]. In this way, it sought to establish its legitimacy as a research company. The public–private partnerships focused on fundamental and applied scientific questions such as *Phytophthora* resistance and plant development genes [27]. Solynta was also one of the initiators of the Holland Innovative Potato (HIP) project. HIP covers fundamental research regarding potato agronomy, phytopathology and secondary metabolism. It is co-funded by private and public partners who all gain access to the results. Very recently, in 2021, Solynta started collaborating with Averis Seeds BV, a Dutch company specialized in the breeding and processing of starch potatoes. The two companies joined forces with the aim to breed hybrid processing potatoes with high levels of starch, adapted to cultivation close to the processing factories.

Today, hybrid breeding at Solynta has reached a point where it is a viable breeding method, and the development of hybrid varieties for commercialization is well underway [28,29]. So far, the breeding program of Solynta focused on traits deemed to be of general importance, such as uniformity, tuber yield and processing quality. In the future, there may be divergent breeding priorities for specific agroecological zones. However, breeding, specifically for poorly developed niche markets, such as hybrid potatoes adapted to tropical lowlands, is considered unrealistic from a business perspective due to low returns and high risks [7]. Therefore, Solynta is aiming at public–private partnerships that may offer ways to organize the resources needed to address food security and climate change as major global challenges. 

### Hybrid Potato as A System Innovation

The start of a hybrid breeding program based on self-compatible diploid potatoes is described both by outside observers and the Solynta team itself as a revolutionary ‘paradigm shift’, involving the transformation of breeding, seed and cultivation as three interrelated systems [4,5,30]. Firstly, it redefines the traditional and time-consuming tetraploid potato breeding system. The diploid potato can now be rapidly improved through targeted breeding, combining favorable traits from different parental lines, thus transforming breeding into a more controlled and predictive endeavor. Secondly, it changes the potato seed system, affecting the use, availability and production of seed. It is now possible to grow potato from tiny hybrid seeds rather than bulky seed tubers, thus offering farmers a source of clean, disease-free starting material that is easy to transport and can be stored for a long period of time. Whereas conventional seed tubers are slowly multiplied via the vegetative system, a single diploid potato plant produces thousands of seeds per season. Thirdly, the availability of true seed as starting material alters the potato cultivation system. Farmers can now choose to grow potatoes from hybrid potato seed, greenhouse-grown seedlings or first-generation seed tubers [23]. Cultivation from seed (by direct sowing) or seedling transplants may transform potato into a horticulture crop [31], but first, the feasibility of these different options will have to be established in field studies, testing cropping systems, agronomy and farm economics.

## 4. From Promising to Responsible Innovation

With its game-changing potential, the hybrid potato is regarded as a highly promising future contribution to the policy goals of global food security and sustainability. On its website, Solynta presents itself as a company using hybrid potato breeding to grow a more sustainable future. By unlocking the true potential of potato, the company is especially ‘determined to drive sustainable improvements in world food security’ [32]. However, moving from promises in relation to urgent and global societal issues to actions that make a difference at the local level is a challenging task [33]. A task that cannot be fulfilled with technological solutions alone. System innovation involves a complex interplay of (seed) technological, agronomic, commercial, social, cultural and political factors and developments, also including sensitive issues of variety registration, certification and protection.

The future of hybrid potato will thus be shaped by diverse and dynamic system conditions. Inspired by the approach of RRI, we wanted to attain in our POTAREI project a deeper understanding of these conditions through a process of mutual learning [34]. RRI has been conceptualized in terms of four process dimensions that are important in stimulating mutual learning [35]. Anticipation aims at a more detailed understanding of the complex and multifaceted nature of the world in which future innovations should find a place. Inclusion requires engagement with a broad diversity of actors from these worlds. Reflexivity demands the actors engaged critically examine the needs, interests, issues and values at stake in processes of innovation. Responsiveness implies a willingness to take sincerely into account societal purposes, contexts, tensions, needs and values in shaping innovation. In Box 2, we have indicated how we have translated these different process dimensions in a series of activities that have structured and informed our project. In the following, we will describe in more detail the steps that we have taken, the problems that we have addressed and the major findings emerging from our responsible innovation project journey, inspired and stimulated by the developments in hybrid potato breeding at Solynta and elsewhere.

Box 2Four process dimensions of responsible innovation in hybrid potato breeding.Anticipation aims at a more detailed understanding of the complex and multifaceted nature of the world in which emerging innovations should find a place. Accordingly, we explored in our project a range of possible futures for hybrid potato breeding in a global context.Inclusion requires engagement with a broad diversity of actors from these worlds. In thinking about hybrid potato futures, we therefore involved key players from the Dutch and international potato sector, and various representatives from knowledge institutions, civil society and policy making with an interest in these futures. (We also worked with a so-called ‘valorization panel’: an advisory board consisting of a diverse group of stake-holders and experts active in different domains relevant to our project).Reflexivity demands the actors engaged critically examine the needs, interests, issues and values at stake in processes of innovation. We invited the actors involved to reflect on different system conditions for potato breeding, cultivation and marketing and how these conditions and the emergence of hybrid potato breeding may interact.Responsiveness implies a willingness to sincerely take into account societal purposes, contexts, tensions, needs and values in shaping innovation. In this context, we invited actors to address the conditions and policies needed for a future in which hybrid potato breeding can effectively serve food security and other relevant SDGs.

## 5. Mapping Expectations and Building Future Scenarios

One of the core activities in our POTAREI project was an interactive scenario-building exercise in which we explored how the future of hybrid potato might be shaped by different and changing system conditions, with different implications for food security and sustainability on a global level; this section mostly relies on ref. [36]. We started this process in 2016, when hybrid potato breeding was at an early stage of development and its innovation trajectory was still open and uncertain. In such a context, a more detailed understanding of the ensuing innovation dynamics can be gained by looking at the expectations of different stakeholders about the future of this innovation. Therefore, we started our exploration by interviewing a diversity of stakeholders within and outside the Dutch potato sector—breeders, seed potato farmers, trading houses, civil society organizations, experts and policymakers—in order to map their expectations about the impact of hybrid potato, both on the sector and society at large. A systematic analysis of these expectations helped us better understand what was at stake in diverging claims about the future, which opportunities and hurdles were perceived, and what would drive or inhibit stakeholders from acting on particular expectations. On the one hand, this first step in our exploration was about immersing ourselves as a multidisciplinary project team in the realities, practices and expectations within the field; on the other hand, it was about putting key actors into a process of responsible innovation.

By mapping our interview findings along two dimensions—the expected impact of hybrid breeding on the potato sector and the expected impact on society—we found three clusters of actors’ expectations, with each cluster including a wide variety of stakeholders [30]. We found a cluster of actors who shared the expectation that hybrid potato breeding will meet unsurmountable technical and societal barriers and thus will bring no significant change at all. The second and largest cluster included actors showing higher expectations of the technical feasibility of hybrid breeding without expecting more than modest effects on the sector and society at large. Finally, there was a third cluster of actors describing hybrid potato breeding as a game changer that may bring a technological solution to food security and sustainability as urgent societal problems, with potentially disruptive effects for the sector.

Central to the low or moderate expectations about the impact of hybrid breeding was the idea that the potato sector itself is highly unchangeable. Actors in the first two clusters considered innovation mostly in light of established sectoral structures. They perceived these structures as barriers that would be difficult to overcome. Representatives of large potato processing companies, for example, pointed out that any innovation in breeding should meet the tight-fitting requirements of the processing industry. More in general, interviewees believed that the introduction of new potato varieties is marked by a strong path dependency and also repeatedly mentioned, in this context, the relatively conservative nature of the Dutch potato sector. To the extent that hybrid breeding was deemed technically feasible, it was thus expected to lead to innovation that fits well with established farming and processing systems, and it was seen as unlikely to bring broader sectoral or societal changes.

Actors in the third cluster, on the other hand, were more optimistic about the potential for change. They did not take the existing structures of the potato sector as their starting point, but emphasized urgent sector-wide problems, with implications for food security and sustainability as global challenges and hybrid breeding as a highly promising technology to overcome various constraining obstacles. Nevertheless, these interviewees were also keenly aware of the challenges that such changes would entail.

### Co-Construction and Discussion of Scenarios

In the divergent claims about the future of hybrid potato, we find different perceptions of the ways in which new technological options and existing system conditions may shape each other. A deeper understanding of this innovation dynamics is crucial for the aim of responsible innovation. This is where scenarios can help, not as predictions of the future but as a way to more systematically explore how multiple versions of the future might unfold. Accordingly, we brought together different stakeholders from the Dutch potato sector and beyond (including some of the interviewees and members of our valorization panel) in a joint scenario-building exercise in order to bring about a process of mutual learning about possible and desirable hybrid potato futures [34].

To structure this stakeholder dialogue, we first developed three basic storylines based on insights from the literature and our stakeholder interviews, showing how the embedding of hybrid potato in future food systems may go in different directions. Each storyline dealt with food security and sustainability as grand societal challenges in a different way, leading to specific internal tensions, with both winners and losers. As we aimed to contribute to optimal conditions for the responsible development of hybrid breeding, we took the successful introduction of hybrid potato seed on the (global) market as a starting point for our scenario-building exercise.

The three storylines were, as basic scenario frameworks, further elaborated and discussed in two successive stakeholder workshops. In the first workshop, we invited the participants to look forward 25 years and to think out of the box, thus stimulating their imagination and broadening their perspective beyond personal or sectoral interests. Based on the outcomes of the first workshop, we elaborated the narrative within each scenario more fully (see Box 3). The resulting scenarios served as a starting point for a second workshop, focusing on questions such as: Do these scenarios make sense, and what else might happen in each of them? The scenarios were also discussed by the participants in terms of plausibility and desirability, including questions about how to strengthen the positive aspects and how to mitigate the negative aspects of each scenario.

Box 3Three hybrid potato future scenarios [34,36].The first scenario—Global Duopoly—is in line with the current dominant trend in agriculture towards increasing scale and intensification of farming. In this scenario, hybrid breeding is in the hands of two large companies, using true potato seeds to grow uniform potatoes in bulk. A second scenario—Sustainable and High-Tech—is inspired by the ecomodernist vision of sustainability. In this scenario, hybrid breeding is part of a highly efficient, technology-driven farming system, guided by changing environmental and climate policy requirements. In a third scenario—Diversified Markets—hybrid potato breeding is globally supported by the public availability of homozygous potato parental lines, enabling breeders all over the world to develop a broad range of hybrid potato varieties suited to local conditions.

The scenarios show that hybrid potato breeding can play a role in very different futures, with various impacts on food security, short and long-term ecological sustainability, and the economic position of different stakeholders in the potato sector. Clearly, different future development paths are possible, but in the workshop discussions, not every path was seen as equally plausible and self-evident. In this respect, the scenario exercise revealed a remarkable tension between the plausibility and desirability of the futures discussed. Even though the Diversified Markets scenario was marked as an attractive future by a significant number of participants, the Global Duopoly scenario strongly figured as the most plausible one.

Apart from the pros and cons of the different scenarios, we also asked the participants to draw lessons that could help formulate more robust, scenario-transcending conditions and strategies for responsible innovation in (hybrid) potato breeding. A few core issues emerged from the workshop discussions as overarching lessons, connecting the future of hybrid potato more strongly with the public interests of global food security and ecological sustainability [34]. Full access to genetic resources, knowledge and technologies was seen as a precondition for an innovative market, ensuring a broad genetic diversity in commercially available potato varieties and promoting a diversified market approach. This requires continuous public and private investments in knowledge development, relating to (hybrid) breeding, cultivation and the organization of potato value chains within different global contexts, which should also stimulate the development of specific know-how about local production conditions and about breeding locally adapted varieties.

## 6. The Governance of Genetic Resources in (Hybrid) Potato Breeding

In light of the lessons from our scenario exercise, we defined the governance of genetic resources as a major subject that would deserve a more detailed study from a so-called ‘commons’ perspective; this section mostly relies on ref. [37]. Commons refer to shared resources that are governed by communities of users according to rules that make these resources available and accessible in sustainable ways. In an elaborate series of case studies, Ostrom and colleagues have elicited the nature and importance of institutional arrangements in maintaining resources as commons [38]. From this institutional perspective, plant genetic resources—whether seeds or genetic information—have been extensively discussed as commons. In particular, plant genetic resources for food and agriculture (PGRFA) have become a subject of global interest and international legal regulation, with the aim to increase their sustainable use and conservation and to promote equitable benefit sharing [39].

Moreover, PGRFA have been characterized in this literature as a ‘new commons’ because of their partially human-made nature as a result of a long history of agriculture-related practices of seed selection, cross-breeding and conservation [40]. In conceptualizing the commons, we should, therefore, not only focus on institutional conditions, but also on how the availability and accessibility of resources are mediated by human intervention and technological developments. Aiming at a more detailed understanding of the relation between the commons and innovation, we decided to investigate how the governance of potato genetic resources and the emergence of hybrid potato breeding interact, with a focus on practices of potato breeding in the Netherlands.

### 6.1. Potato Genetic Resources Governed as Commons and Meditated by Technological Innovation

In the Netherlands, potato genetic resources are freely accessible, although bounded by contract, through the Dutch gene bank, storing over 2800 potato accessions. Under the In Trust Agreements on the acquisition and distribution of germplasm by institutes of the Consultative Group for International Agricultural Research (CGIAR), users around the world can also freely access genetic material held in international gene banks [41]. Gene banks are therefore considered key institutions in governing plant genetic resources, ensuring the wide availability and accessibility of these resources as commons by a set of practices, rules and norms [42]. Moreover, genetic resources are also made available and accessible through more informal practices of sharing. One example is the collaboration between trading houses and collectives of farmer breeders as a rather unique feature of the Dutch potato breeding sector, whereby farmer breeders help breeding companies in selecting plants in the field that are suitable for further breeding [43].

As we observed in our study, the possibility to use potato genetic material for breeding purposes does not only depend on the institutional availability and accessibility of these materials, but increasingly also on techno-scientific knowledge and skills that enable breeders to unlock genetic material as a new resource. Breeding companies now extensively use genetic marker technology for identifying plantlets with favorable genes [44]. To stimulate the development and sharing of knowledge, skills and materials that may help to unlock potato genetic resources, the Dutch government has been funding collaboration between universities and trading houses under the condition that all research participants will continue to have access to the newly unlocked genetic material, thereby ensuring the availability and accessibility of this material as commons.

With the rise of genomics as a new innovative field of research, the Dutch government likewise funded public–private partnerships, among which a Centre for Biosystems Genomics that was involved in the international Potato Genome Sequencing Consortium, providing a public platform that makes potato sequence data available and accessible to the scientific and breeding community at large [45]. In the context of genomics innovation, however, the support of public–private partnerships by the Dutch government went hand in hand with the promotion of intellectual property rights by patenting, reflecting a more general and global trend in plant biotechnology, genomics and commercially driven plant breeding [46,47,48]. Thus, we see tension arising between unlocking and appropriating plant genetic resources, raising concerns and debates about how to balance sharing for the common good with a propensity to protect [49,50].

### 6.2. The Rise of Hybrid Potato Breeding and its Implications for the Governance of Genetic Resources

Against this background, we explored in our study the implications of hybrid potato breeding for the governance of genetic resources. As a new and emerging innovation, hybrid breeding may help to further unlock the large genetic variety of potato as a resource. Conversely, its development is also highly dependent on the availability and accessibility of potato genetic resources as commons. Indeed, the public availability of diploid germplasm and a self-compatibility restorer gene paved the way for Solynta’s innovation journey. Key steps in hybrid potato development have also been enabled by skills, knowledge and (gene) technologies that are widely available, for example, molecular DNA ‘markers’. In the context of hybrid breeding, such markers can be used to test the level of homozygosity in inbred parental lines and also to predict more accurately the breeding value of these lines [4]. Moreover, by so-called marker-assisted introgression breeding, publicly available favorable traits can be introduced into these lines, for example, the *Phytophthora* resistance genes that were stacked by Solynta in their potato parental lines [12]. As noted by an international group of public and private sector scientists, by “reinventing the potato crop at the diploid level”, hybrid potato breeding could “take full advantage of the modern genetics and genomics tools available to improve gain from selection” (2: p.2). Thus, to harness the power, precision and speed of hybrid breeding technology, the wide availability and accessibility of genetic resources as commons have been critical.

In our study, we also sought to understand how the introduction of hybrid breeding as innovation may affect the governance of potato genetic resources. As hybrid potato breeding is still in its infancy, our discussion was mostly forward-looking and explorative. At first sight, the potential implications of hybrid potato breeding for the governance of genetic resources seem to be fairly limited. It may, first of all, lead to a further unlocking of potato genetic resources that will become available in improved varieties. In one respect, however, its introduction entails a crucially important institutional change in the governance of potato genetic resources. Companies involved in hybrid breeding have to make huge investments in the development of inbred parental lines as the main building blocks for their own business. Therefore, companies generally consider these lines as their main economic asset and take strict measures to ensure that access to their parental lines is restricted. Indeed, the shift from conventional to hybrid breeding, as in maize and other crops, has historically been associated with increasing (natural) protection and corporate appropriation of (genetic) knowledge and with commercial marketing of seeds around the world [48,51].

What future developments can we envisage for hybrid potato breeding in this context, and what could be the implications of these developments for the governance of genetic resources? As hybrid breeding is further getting off the ground, potato breeders may compete more and more on the basis of proprietary parental lines, considering their lines as a trade secret and probably also applying for patents on genetic traits that may further limit access to genetic materials [8]. This may result in high entry barriers for newcomers in the field. As the ownership of such precious parental lines has high economic value, hybrid potato breeding might lead to a series of mergers and acquisitions in the potato value chain, akin to the mergers and acquisitions previously witnessed in the seed sector of other crops [48,52]. Pushing the argument further, this could create a situation where a handful of corporate seed players gains sufficient size and capital needed to privately undertake the large-scale research efforts in potato breeding that currently still require commons-based collaboration and government coordination.

The tension we noted above between unlocking and appropriating plant genetic resources becomes visible here in a tension between commons-based and corporate-based modes of governing genetic resources, which we also encountered already in the different futures resulting from our scenario-building exercise. The first mode of governance is driven by the sharing of knowledge and materials and by collaboration to strengthen the knowledge base for (hybrid) potato breeding. The second mode is driven by market-based forms of research and protection of potato genetic resources and by corporate consolidation of (hybrid) potato breeding. As we will discuss below, this tension is also raising questions about how these different governance modes match with the various goals that are pursued in breeding, especially in regard to the international SDGs.

## 7. Hybrid Potato Breeding and Sustainable Development Goals

In the foregoing, we have described how we explored the (potential) interactions between hybrid potato as a new innovation and the wider system conditions in which this innovation must gain a foothold, taking Solynta’s innovation journey as our starting point. As the company started to consider the potential impact of hybrid potato on global food security, our POTAREI team also felt incentivized to more deliberately explore how hybrid potato could make a difference in the international development context of Sub-Saharan Africa. The emphasis on Africa was largely motivated by the large and persistent yield gap of potato in this continent due to a lack of good seed and a range of pathogens that resource-poor smallholder farmers struggle to control [16,17,18,19]. Under such harsh socio-ecological conditions, hybrid potato may offer a significant contribution to food security and related SDGs, whereas in China and India, as prominent potato producers, the socio-ecological and institutional conditions are more favorable, and national initiatives have already been initiated to develop domestic hybrid potato breeding programs [53,54], see also Section 7.1.

A first step in our exploration of these issues was the organization of a workshop with participants and presenters from National Agricultural Research Institutes, universities, international seed companies, the International Potato Center (CIP) and governmental organizations, covering a wide variety of African countries. The main aim of this workshop was to better understand how potato cultivation in East Africa could possibly be improved by the introduction of hybrid true potato seed. We conceived a program in which technological innovation was considered in a societal context of mutually affecting ‘worlds’, with the world of African potato cultivation and breeding at the center of our scheme. We also prepared a background paper about potato in Africa, with observations from the literature, that was included in our final workshop report [55].

One year later, in the fall of 2020, we organized an international conference about Potato Futures: Impact of hybrid varieties, with speakers from private industries, universities and research centers, donor and non-governmental organizations and policy-making institutions. (The organization of this conference was supported by the Netherlands Food Partnership: www.nlfoodpartnership.com (accessed on 19 April 2023)) The conference was followed mostly online by more than 150 participants from all over the world. The future of hybrid potato was discussed in two parallel program tracks, one focusing on the Dutch agro-industrial potato sector, the other on the international development context of Sub-Saharan Africa. There was a broad agreement about the great potential of hybrid potato for future food security on a global scale, but how this innovation relates to prevailing system conditions, both in the agro-industrial and international development context, was an important issue of debate [56]. To complete our POTAREI mission and take stock of what we have learned, we edited a volume covering this debate, with contributions inspired by the international Potato Futures conference [57].

### 7.1. The Future of Hybrid Potato in the Agro-Industrial Context

The conference discussions about hybrid potato in the agro-industrial context mostly focused on the Netherlands as a vibrant center in the international potato world, highly involved in every aspect of the potato value chain [58]. This particular Dutch perspective is also relevant for a broader understanding of the challenges and implications of hybrid breeding in various market contexts; this section mostly relies on ref. [59]. Hybrid breeding will allow breeders to respond much quicker to new market demands, with a significantly higher variety turn-over than in conventional potato breeding. The potential of hybrid potato has now been generally recognized, and fundamental hybrid breeding programs have been launched in the largest potato-producing countries, China, India and the USA [53,54,60]. However, building new hybrid breeding programs from scratch requires long-term efforts and high commercial investments [7]. Before we can expect improved hybrid potato varieties to reach the agro-industrial market, these varieties will first have to meet current, high-level and tight-fitting retail and potato processing standards.

Hybrid potato breeding may also provide new opportunities for organic farming, with its specific growing requirements. In this sector, potato is one of the most difficult crops to cultivate as proper means are lacking to control the many pathogens threatening the crop. By quickly stacking resistance genes, hybrid varieties can be generated that contain resistances badly needed for organic farming, notably against late blight [61]. However, seed sovereignty is very important in organic farming, and farmers may be reluctant to adopt hybrids that limit their autonomy by making them dependent on seed companies [62]. Yet, within the organic sector and between different countries, there are markedly different opinions on this point, ranging from principally avoiding hybrids to embracing them as an opportunity.

The availability of uniform and virtually disease-free true seed as starting material for potato cultivation is another important benefit accruing from hybrid breeding. Farmers may now grow potatoes from hybrid potato seed, seedlings, mini-tubers or first-generation seed potato tubers. Growing potatoes directly from seed, transplants or mini-tubers will require, however, quite a different agronomy, and it remains to be seen what will be the most appropriate cultivation models in different agroecological conditions. The use of hybrid potato starting material will also imply, for each of these cultivation systems, a significant shift in the creation of added value from the conventional seed tuber grower to the breeding/seed company. Moreover, as true potato seed can reach areas that currently do not have access to high-quality seed potato tubers, hybrid seed may also create opportunities for these companies to expand their export market. The emergence of hybrid potato may thus transform as well as strengthen established business models in the agro-industrial potato sector.

An important policy issue in this context is the need for adjustment of global regulatory frameworks to the use and trade of hybrid true potato seed [29,63]. The official protocol for testing Distinctness, Uniformity and Stability (DUS) of tuber-based potato varieties has to be adapted to seed-based hybrid varieties. Examination of the Value for Cultivation and Use (VCU) of hybrid potato varieties also requires a different approach, as well as seed quality control and certification. Although a shift towards the export of true potato seed could reduce the need for phytosanitary measures, many countries have not yet opened their borders for the import of hybrid seed. An important issue to consider in this context is the impact of strict and formal seed regulations on the availability and use of hybrid potato seed in the informal seed sector, which remains of overwhelming importance to smallholders in large parts of the world. Indeed, with a view to global food security, there is much to be gained by lowering regulatory thresholds for smallholders and making the formal system more inclusive [64].

### 7.2. The Future of Hybrid Potato in the International Development Context

A key question for our project to address from the perspective of responsible innovation, is how the global challenges of food security, agrobiodiversity and climate change can be made key drivers in the development of (hybrid) potato breeding. Our workshop and conference activities offered an excellent opportunity to further explore this question, in particular by focusing on the African international development context where the issues of food security and climate change are most challenging; this section mostly relies on ref. [59]. In this context, it is especially important to take into account the dominance of informal seed systems: potato seed tubers are mostly farm-saved, exchanged within farmers’ communities or marketed locally, thus sustaining the livelihood of millions of resource-poor smallholder farmers. There is a clear difference and distance between this informal seed system and the formal agro-industrial system. In the international development context, therefore, switching from local farmers’ varieties to commercial hybrid varieties represents a major shift. In discussing this shift, two issues are to be considered as a main agenda for debate (see Box 4).

Box 4Our main agenda for debate [65].Hybrid breeding will be an important driver for rapid variety development, creating new added value, both commercially and from a societal point of view. This raises questions about the needs to which this variety development should respond. How to connect commercial interests and values with innovation that contributes to the common good, responding to food security, sustainability and climate change as major societal challenges expressed in the global sustainable development goals?Hybrid varieties will become available as true potato seeds that can be propagated, stored and transported as a commercial (or public) source of clean and high-quality planting material, but with less growth vigor than the commonly used seed tubers. This creates new opportunities and challenges for the organization of potato value chains, with cultivation systems that may vary from direct sowing by farmers, to using plantlets or seedling tubers. How to make hybrid true potato seed available and accessible to smallholder farmers in the form of useful planting material?

A crucial requirement for an effective and inclusive introduction of hybrid potato in the African international development context is the mutual alignment between the characteristics of this innovation and the diverse needs, preferences and realities of African potato farmers. How do we accommodate the wide diversity of farmers and farming systems in Sub-Saharan Africa? On this point, we find, in the contributions to our Potato Futures conference volume, two different perspectives on the kind of changes and improvements that hybrid potato could or should bring in the international development context. On the one hand, hybrid potato breeding is discussed as a game changer that may cause disruptive change, with hybrid potato as a driver of potato sector transformation, aiming at the creation of commercial hybrid seed value chains. From this perspective, regionally prevailing farming practices, cultures and traditions are perceived as barriers that have to be surmounted [66,67]. On the other hand, we find pleas for an approach in which breeding innovations and system adaptations are tailored towards the needs and capabilities of different farm household types. Starting points, from this perspective, are the typical agroecological and socioeconomic farming conditions to which hybrid varieties and seed value chains should be attuned [68].

In viewing the future of hybrid potato, we thus find a tension between transformation and integration as potentially conflicting approaches [59]. The more radical the transformation—with hybrid seed based on an agro-industrial variety portfolio, requiring labor- and capital-intensive crop management—the more obstacles will be created to the adoption of this seed by smallholder farmers. The more radical the integration—with hybrid varieties tailored to the practices, preferences and needs of smallholder farmers—the less likely that companies will see a business model for dedicated seed development and production. A crucially important question raised by this tension is how to find a balance between these two approaches, with the aim of serving global food security in inclusive and sustainable ways.

## 8. Responsible Innovation Strategies in (Hybrid) Potato Breeding

Hybrid potato breeding may contribute to the productivity, sustainability and diversity of potato production systems, but this will not come about spontaneously. By no means is it a foregone conclusion that these promises will actually be fulfilled. It was the aim of our POTAREI project to better understand the complex and uncertain ways in which hybrid potato breeding and different system conditions may shape each other, and also to consider the tensions that may arise between these conditions and the goals and promises that breeders have in mind for the future of hybrid potato. We introduced responsible innovation as an approach that fosters mutual learning, involving a broad variety of stakeholders early in the innovation process, with the aim to direct innovation to major societal challenges in the midst of a complex and uncertain world. Inspired by this approach, we chose to focus on the potential contribution of hybrid breeding to SDGs in the international development context, with the conditions and needs of smallholder farmers as an important pointer for guiding hybrid breeding in desirable directions, particularly with regard to the challenges of food security, agrobiodiversity and climate change. 

As global food security policies are mostly centered around the commercialization of agriculture and ‘market-led technology adoption’ [69], there is a crucial role to play for breeding companies in targeting variety development to meet smallholders’ needs. One of the contributions to our Potato Futures conference volume explores how Dutch potato breeding companies perceive their responsibility with regard to food security and related SDGs [70]. It shows that firms indeed keep these goals in focus from a Corporate Social Responsibility (CSR) point of view. (Hybrid) variety development and global seed trade are seen by these companies as core activities that give substance to their CSR efforts. Resource-poor smallholder farmers may benefit in this view from improved, commercially produced seed through a ‘trickle down’ process from the formal sector to informal community networks.

However, as we already noted in the introduction, among scholars studying seed interventions in the international development context, there is a broad agreement that without a more in-depth understanding of farmers’ diverse conditions and needs, such interventions are often doomed to fail. Moreover, as observed in another contribution to our conference volume, without substantial market demand, we cannot expect that traits that are specifically relevant to smallholder farmers will receive much attention within commercial (hybrid) breeding programs [19]. Indeed, the question of how to combine business opportunities for breeders with variety development that serves global food security was one of the key issues mentioned at the international Potato Futures conference.

### Need for Collective Action

In the previous section on hybrid potato breeding and SDGs, we concluded that seed interventions require a delicate balancing act, aptly phrased in the literature as “introducing viable and scalable innovations, while meeting a wide variety of smallholder needs and requirements” [69]. A recurrent argument in the contributions to our conference volume is the need for partnerships and collaboration in order to enable this balancing act, as neither the private, nor the public sector can fully harness the potential of hybrid breeding alone. One of these contributions focuses on the question of how this collaboration can best be organized in order to facilitate the development and dissemination of potato varieties that are suited to the specific conditions and needs of smallholder farmers [71]. It explores—from a commons perspective—four models for institutionalizing public–private partnerships and assesses the potential of each model to collectively target hybrid breeding at smallholder farmers in response to the challenges of poverty, food security and climate change (see Box 5). The authors silently hope that a better understanding of the strengths and weaknesses of these different models may serve as a starting point for the public and private sectors to come together and discuss how they can combine their forces for the benefit of smallholder farmers around the world (see also: [2]).

Box 5Four models of public–private partnerships in hybrid potato breeding [71].In a charity model, the breeding of hybrid diploid potatoes is concentrated in the private sector, while the dissemination of starting material to smallholder farmers is taken up by the public sector and non-governmental organizations (NGOs).In a pre-breeding research model, the early stages of hybrid potato breeding are concentrated in the (inter)national public sector, while the further selection and dissemination of hybrid potato to smallholder farmers are taken up by the private sector.In a breeding consortium model, parental lines are developed by the private sector and are then shared with the public sector under the restrictive mandate to use these exclusively to develop varieties with traits that are specifically relevant to smallholder farmers.In a project model, public and private institutes collaborate on an ad hoc basis to tackle specific breeding challenges when there is a shared interest, which can be understood as a form of project-based work and may apply to any of the three other models of collaboration.

In the foregoing, we discussed the implications of hybrid breeding for the governance of potato genetic resources, noting the prospect of a rising tension between corporate-based and commons-based modes of governance that may both shape the future of hybrid potato breeding. Our discussion of the future of hybrid potato in the international development context made clear that, in order to fully reap the benefits of this innovation, we cannot leave the development of hybrid potato breeding to corporate-based governance only. If we would like hybrid breeding to be responsive to the major societal challenges addressed by the SDGs, we also need to think about scenarios in which potato genetic resources remain to be governed as commons.

Therefore, we called for the establishment of a breeding consortium that can realize the potential of hybrid potato breeding for food security by making this innovation widely accessible [72]. We argued that access to inbred potato parental lines, as a crucial genetic resource for hybrid breeding, is currently restricted to breeding companies that are willing and able to make long-term investments in the development of these lines, while public breeding research centers, as mainly project-funded organizations, do not have the capacity to develop full-fledged hybrid breeding programs. The limited accessibility of parental lines as a private asset will necessarily constrain the potential contribution of hybrid potato to global food security, inasmuch as a focus on commercialization and profit-seeking inherently favors the development of varieties that serve high-value markets rather than the specific needs of smallholder farmers. What we need, then, is an institutional intervention, balancing different interests and values, with the aim of serving global food security by exclusively targeting smallholder farmers. Our proposal includes partnerships that strengthen the role of public breeding, stimulate the private sector to fulfill its mandate of corporate social responsibility and may create incentives for farmers to both share in situ genetic resources and participate in breeding (see Box 6). Thus, we have been seeking, in our POTAREI project, to more specifically define institutional conditions that meet a true commitment to responsible innovation in hybrid potato breeding.

Box 6A hybrid potato breeding consortium serving global food security [72].The consortium that we propose may be based on different partnership models, with benefit sharing as the basic principle. In one of these models, private breeding companies participate in the consortium by sharing their potato parental lines, thus making available these lines to the participating public institutions, which can use them for hybrid variety development that serves smallholder farmers’ needs. While the breeding consortium ensures the accessibility of hybrid planting material exclusively for smallholder farmers, the participating companies are allowed in this model to market hybrid varieties acquired from the public consortium partners. In another alternative model, public breeding institutions participating in the consortium may direct the generation and selection of hybrid cultivars in view of the needs of resource-poor farmers, by obtaining hybrid seeds from a dedicated program of crosses between potato parental lines, carried out by the participating private companies owning these lines. In this model, the companies are free to market these seeds, while waiving their breeding rights for hybrid planting material that is made accessible by the breeding consortium exclusively for smallholder farmers.

## 9. Further Outlook and Conclusions

In this article, we showed how the approach of responsible innovation could stimulate mutual learning by including a broad variety of stakeholders and other relevant actors early in the innovation process, enabling them to anticipate potential impacts and also reflect on desirable conditions and outcomes of innovation. This process enabled us—as a multidisciplinary research team—to develop a broader understanding of the innovation dynamics shaping the future of hybrid potato and also to address the tensions between the predominantly commercial innovation dynamics of hybrid breeding and the global challenges of food security, agrobiodiversity and climate change. The workshop and conference discussions that we helped organize, further stimulated thinking among the participants about strategies that could possibly bridge these tensions. Inspired by these insights, we have published proposals for institutional reform of the international potato breeding playing field, with the aim to enhance the responsiveness of hybrid potato breeding to the diverse needs and conditions of smallholder farmers. What can we conclude about the possibilities and opportunities of such institutional interventions in support of responsible innovation, especially in light of the experiences of Solynta as a pioneer in hybrid potato breeding and partner in our POTAREI project?

### 9.1. Stretching the Limits

From the perspective of responsible innovation, one might expect that the anticipatory and inclusive processes of learning inspired by our POTAREI project could also give direction to Solynta’s innovation journey. As we have seen, however, Solynta, as a company, and POTAREI, as a multidisciplinary project, have mostly gone their separate ways. For Solynta, the prime rationale of its activities is the development of a radically new diploid hybrid breeding program, with the aim to generate hybrid varieties that can match or even outperform the tetraploid varieties that are currently dominating commercial and agro-industrial potato markets. The opportunities for Solynta to respond to the challenges and ambitions articulated in our POTAREI project appeared to be limited, both by the current state of the art of its nascent breeding program and by the sheer necessity of breeding varieties that offer prospects for return on investment. Indeed, the tension between system conditions in which commercial plant breeding has to operate and wider aspirations to respond to global challenges, also manifests itself in the relation between hybrid potato innovation dynamics on the one hand, and the aims of responsible innovation on the other. This is even more true for young companies like Solynta, which may have to deal with a race against the investors’ clock.

Nevertheless, Solynta has taken the initiative to explore the potential of hybrid potato in a series of experimental trials with hybrid varieties in different African countries. The hybrids that are tested under various agroecological conditions are selected by the company from its current breeding innovation pipeline. In this respect, the company follows a strategy similar to that of other globally operating potato breeding firms, which in their breeding programs hardly focus on the particular needs of African farmers but introduce best-bet varieties from their portfolios in those countries where there is a market demand [19]. As we learned from our exploration of hybrid potato prospects in the international development context, in the absence of an immediate business case, there is a need for public–private collaboration to facilitate the development and dissemination of potato varieties that are suited to the specific conditions and needs of smallholder farmers (see also: [7]). To further strengthen Solynta’s pilot activities and partnerships in Sub-Saharan Africa, a joint program has been proposed, seeking to bring together local public and private stakeholders in a new collaborative platform initiative called Sepia [73]. With branches to be established in several African countries, this initiative aims to facilitate the introduction and promotion of hybrid potato on a wider scale. Initially, the Sepia Foundation will introduce readily available hybrids to commercial farmers. In five to ten years, as the market starts to grow, there could be more emphasis on the dedicated breeding of new varieties, responding to the needs of a greater diversity of farmers, including smallholders. As a collectively organized institutional intervention, the Sepia initiative might thus stretch the limits of the innovation dynamics in hybrid potato breeding.

### 9.2. Towards a More Responsive Plant Breeding Playing Field

In the introduction of this article, we introduced responsible innovation as an approach that can help plant breeders consider the wider societal implications of their work, and that resonates with a more general debate in the literature about the role of plant breeding in responding to the global challenges of food security, agrobiodiversity and climate change. What conclusions can we draw from our own experiences and findings about this more general debate? The tensions that we have noted in our exploration of hybrid potato futures clearly illustrate the need for an inclusive and diverse plant breeding playing field, as commercial breeding in an agro-industrial system context cannot sufficiently cater to the diverse needs and conditions of smallholder farmers in an international development context [8].

Such an inclusive and diverse playing field implies a move beyond the formal agro-industrial system context toward the informal worlds of resource-poor smallholder farmers working in highly diverse low-input agroecological farming conditions. Collaboration with farmers in plant breeding has a long history, including a well-functioning model of farmer participation in Dutch potato breeding [43,74]. Since the 1990’s, participatory breeding has become a widely adopted model in the international development context, with the aim to involve smallholder farmer communities in the development of varieties that are adapted to diverse farmers’ needs, also taking into account the specific and often challenging agroecological conditions in which farmers operate [75]. Participatory breeding is further advocated as a strategy that can support smallholder farmers in maintaining and improving crop genetic diversity, emphasizing seed as a common good [76,77]. A community-based and agroecological systems approach could thus enable plant breeders to more effectively respond to the need for cultivars that match diverse and changing socio-cultural, ecological and climatic conditions in an international development context [78].

However, there is a stark unevenness in the positioning of different breeding efforts in the international plant breeding playing field. While commercial breeding is firmly entrenched in a highly structured and globally supported formal agro-industrial system, community-based and agroecological systems approaches are more fragmented in local initiatives, without substantial and sustained public or private support [79,80]. From a responsible innovation perspective, it would be critical for the plant breeding community to initiate a movement towards a more balanced and integrated plant breeding playing field, stimulating potential synergies between differently orientated practices of breeding, and thus enhancing the responsiveness of plant breeding to food security, social justice, agrobiodiversity and climate change as global sustainability challenges.

## Data Availability

No new data were created or processed in preparing this article. Data sharing is not applicable.

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
