# Peer review of "Responsible Innovation in Plant Breeding: The Case of Hybrid Potato Breeding"

_plants, 2023, doi:10.3390/plants12091751_

Round 1

Reviewer 1 Report

Although the manuscript by Stemerding et al. is not a typical research article, the insights provided in this study is valuable and though-provoking for all players involved in hybrid potato breeding. Thus, I think this paper is suitable for publication in Plants. This manuscript pays much attention to the tension between a predominant commercial innovation dynamics in plant breeding and promises to respond the global challenges for food security, agrobiodiversity and climate change, especially in East Africa. To make this manuscript be of more broad interests, I think some other points should also be discussed.

1. The authors should discuss more about the main technical and societal barriers that hybrid potato breeding faces in Europe and American markets. Compared with traditional potato breeding, hybrid potato breeding has many advantages. Whether the potato sectors in these markets are changeable depends on if the breeders can develop diploid hybrid varieties that can match or even outperform the current dominant tetraploid varieties. If yes, how long does this process will take? If not, what are the main obstacles?

2. The recurrent theme of this manuscript is the tension between commercial breeding and the UN Sustainable Development Goals. The authors mainly paid attention to Africa’s potato sector. Actually, the cultivated area of potato in Africa is limited. The potential future of hybrid potato breeding in China and India should also be discussed. They are not only the two largest developing countries, but also the top two potato producing countries. Besides, their cultivation systems are more mature and advanced compared with that in Africa.

3. About the hybrid potato future scenarios, the author wrote “the Global Duopoly scenario strongly figured as the most plausible one”. I think this viewpoint neglected a fact the hybrid breeding in other crops, such as maize and rice, is very mature, and even old. Once the two key obstacles (self-incompatibility and inbreeding depression) that hybrid potato breeding faces were totally overcome, hybrid potato breeding has no essential differences with other crops technically. At now, companies generally consider the parental inbred lines as their main economic asset, and public access of these lines is difficult. However, once the hybrid potatoes were commercialized, other potato breeders can easily develop new and better inbred lines from them by continuous selfing or haploid induction and integration of other beneficial alleles, which has no technical barriers and does not need too much investment. Thus, the Diversified Markets scenario is not only an attractive future but should also a more plausible one.

4. The progress of hybrid potato breeding in other countries, such as China and USA, should also be mentioned and discussed in the manuscript.

Author Response

In response to the first two points mentioned by the Editor, we have included a sentence to the second paragraph of section 6.2. (p.10), referring to the parallel development with maize (and other hybrids), with a reference to Kloppenburg’s documented history of this development. In the introduction of section 7 (p.11), we further explained our decision to concentrate on Sub-Saharan Africa. This explanation also responds to the second comment of reviewer 1.

In response to the first (second) and fourth comment of reviewer 1 we added a section 7.1. (pp. 12-13) about the future of hybrid breeding in the agro-industrial context, explaining the advantages of hybrid breeding and a number of technical and societal barriers in this context. In this discussion, we also refer to developments in China, India and the US as main potato producers.

As to the third comment of reviewer 1 we saw no reasons for revision. We agree that the scenarios (p.8) may be appraised in different ways, but what we reported in the text is the outcome of discussions among stakeholders who participated in our scenario workshops. Reviewer 1 doesn’t see technical barriers for the Diversified Markets scenario, while the scenario discussions related to the broader innovation dynamics shaping the development of hybrid breeding.

Reviewer 2 Report

The article was based on workshops, interviews and conference materials and etc., reviewer can not assess applied methods.

The application of hybrid potato breeding with aims to cover requirements of small and specific growing regions would be applied for organic farming as well. The participatory breeding and small region of variety distribution is very actual, as well as public sector support and availability of all kind of genetic resources.

Author Response

In response to the comment of reviewer 2 we included in the new section 7.1. (p.12) a short discussion of the potential significance of hybrid potato breeding for organic farming.